# Identification of Denatured Biological Tissues Based on Time-Frequency Entropy and Refined Composite Multi-Scale Weighted Permutation Entropy during HIFU Treatment

**DOI:** 10.3390/e21070666

**Published:** 2019-07-08

**Authors:** Bei Liu, Shengyou Qian, Weipeng Hu

**Affiliations:** School of Physics and Electronics, Hunan Normal University, Changsha 410081, China

**Keywords:** HIFU, time-frequency entropy, RCMWPE, ultrasonic scattered echo signals

## Abstract

Identification of denatured biological tissue is crucial to high intensity focused ultrasound (HIFU) treatment. It is not easy for intercepting ultrasonic scattered echo signals from HIFU treatment region. Therefore, this paper employed time-frequency entropy based on generalized S-transform (GST) to intercept ultrasonic echo signals. First, the time-frequency spectra of ultrasonic echo signal is obtained by GST, which is concentrated around the real instantaneous frequency of the signal. Then the time-frequency entropy is calculated based on time-frequency spectra. The experimental results indicate that the time-frequency entropy of ultrasonic echo signal will be abnormally high when ultrasonic signal travels across the boundary between normal region and treatment region in tissues. Ultrasonic scattered echo signals from treatment region can be intercepted by time-frequency entropy. In addition, the refined composite multi-scale weighted permutation entropy (RCMWPE) is proposed to evaluate the complexity of nonlinear time series. Comparing with multi-scale permutation entropy (MPE) and multi-scale weighted permutation entropy (MWPE), RCMWPE not only measures complexity of signal including amplitude information, but also improves the stability and reliability of multi-scale entropy. The RCMWPE and MPE are applied to 300 cases of actual ultrasonic scattered echo signals (including 150 cases in normal status and 150 cases in denatured status). It is found that the RCMWPE and MPE values of denatured tissues are higher than those of the normal tissues. Both RCMWPE and MPE can be used to distinguish normal tissues and denatured tissues. However, there are fewer feature points in the overlap region between RCMWPE of denatured tissues and normal tissues compared with MPE. The intra-class distance and the inter-class distance of RCMWPE are less and greater respectively than MPE. The difference between denatured tissues and normal tissues is more obvious when RCMWPE is used as the characteristic parameter. The results of this study will be helpful to guide doctors to obtain more accurate assessment of treatment effect during HIFU treatment.

## 1. Introduction

High intensity focused ultrasound (HIFU) is a non-invasive modality for conducting high temperature thermal therapy [1,2]. HIFU has the capability to induce biological effects deep into the body by delivering acoustic energy at a distance from the source, and to kill the cancer cells in treatment region [3]. Therefore, it is important to monitor whether the biological tissue in treatment region has been denatured during HIFU treatment. HIFU monitoring includes ultrasound [4,5], magnetic resonance imaging (MRI) [6], and computed tomography (CT) three kinds of monitoring methods [7]. However, CT may damage to human tissue, MRI has poor real-time performance, high price, and requires equipment magnetic compatibility. Ultrasound has become the research focus due to low cost, good real-time performance, and compatibility with hyperthermia devices [8].

However, it is difficult to accurately intercept the ultrasonic scattered echo signals from HIFU treatment region. So far, the ultrasonic scattered echo signals can only be intercepted based on human experience and estimation. Since the temperature of treatment region is higher than that of the normal region in biological tissue. The ultrasonic scattered echo signal from treatment region will exhibit greater attenuation and frequency shift [9,10,11,12,13]. The time-frequency entropy value will be abnormal high when attenuation and frequency change [14,15]. Although the traditional time- frequency analysis methods, for instance, short time Fourier transform (STFT) [16], wavelet transform (WT) [17], and S-transform (ST) [18] are widely applied in many areas, their disadvantages such as low time-frequency resolution, spectral smearing still limit their application [19,20,21]. For better solving those disadvantages, generalized S-transform (GST) on the basis of ST has been proposed by researchers [22]. Inspired by [14,15], time-frequency entropy base on GST is used to intercept ultrasonic scattered echo signals.

Thermal damage of biological tissues can lead to different signal complexity. Lots of nonlinear analysis methods such as Shannon entropy and sample entropy (SE) have been applied to extract features of biological tissues and identify different biological tissues conditions [23,24,25]. In [23], the Shannon entropy of RF ultrasound time series signal was used to assess tissue damage status after HIFU irradiation. In [24], the Shannon entropy of ultrasonic back-scattered signal was employed to detect the severity of fatty liver. In [25], SE of ultrasonic echo signal was employed to identify the variability of infants.

As a nonlinear analysis algorithm, permutation entropy (PE) has the advantages of simple calculation, strong anti-noise ability and robustness. It is widely used in time series complexity analysis [26]. There are two ways to calculate PE, including the uniform embedding and the non-uniform embedding [27]. Comparing with SE and Shannon entropy, multi-scale permutation entropy (MPE) combines multi-scale and permutation entropy to analyze sequence information more efficiently [28,29]. MPE was selected as a feature of ultrasonic scattered echo signals to distinguish whether biological tissues have been denatured during HIFU treatment [30]. However, MPE ignores the amplitude difference between the same permutation patterns and excludes the amplitude information of the time series. In [31], multi-scale weighted permutation entropy (MWPE) was proposed to solve this problem. Meanwhile, as the scale factor increases, the length of the time series becomes shorter, which results in a sudden change for MPE and MWPE [32,33,34]. To solve the problems, inspired by [31,34], multi-scale weighted permutation entropy and refined composite multi-scale entropy are combined to form a new nonlinear analysis algorithm, named refined composite multi-scale weighted permutation entropy (RCMWPE). RCMWPE analyzes time series complexity including amplitude information. Furthermore, RCMWPE averages the probability and weighted permutation entropy with multi-scale factors to reduce the entropy values fluctuation as the scale factor increases, which improves the stability and reliability of multi-scale entropy.

In this paper, time-frequency entropy based on GST is used to intercept ultrasonic scattered echo signals from HIFU treatment region. The intercepted ultrasonic scattered echo signals are analyzed from nonlinear perspective. Considering its stability and reliability, the RCMWPE algorithm is proposed to analyze the difference between normal biological tissues and denatured biological tissues, which can identify whether biological tissues have been denatured and guide doctors to obtain more accurate assessment of treatment effect during HIFU treatment. The uniform embedding is employed in the process of PE calculation. The outline of this paper is as follows: Section 1 is the introduction; Section 2 is the basic theory which includes time-frequency entropy based on GST, algorithm of MPE and RCMWPE; Section 3 includes the experimental system, interception of ultrasonic scattered echo signal based on time-frequency entropy, RCMWPE and MPE of actual ultrasonic scattered echo signals; finally, Section 4 is the conclusion.

## 2. Methods

### 2.1. Time-Frequency Entropy Based on Generalized S-Transform (GST)

Stockwell proposed the S-transform [18] to map time domain signal into the time-frequency domain, the S-transform of signal *x*(*t*) is defined as:
(1)$$S\left(b,f\right)=\int_{-\infty }^{\infty }x\left(t\right)\varpi \left(t-b,f\right)e^{-j2\pi ft}dt$$
(2)$$\varpi \left(t-b,f\right)=\frac{\left|f\right|}{\sqrt{2\pi }}e^{-\frac{f^{2}{\left(t-b\right)}^{2}}{2}}$$
where t is time and b is time shift, f is frequency, ϖ(t−b,f) is the Gaussian window function. Generalized S-transform [22] was proposed by adding an adjustable factor to the Gaussian window function as the Equation (3) shown
(3)$$\varpi \left(t-b,f,\lambda \right)=\frac{\left|\lambda f\right|}{\sqrt{2\pi }}e^{-\frac{\lambda^{2}f^{2}{\left(t-b\right)}^{2}}{2}}$$
Then the Generalized S-transform is obtained as:
(4)$$GST\left(b,f,\lambda \right)=\int_{-\infty }^{\infty }x\left(t\right)\varpi \left(t-b,f,\lambda \right)e^{-j2\pi ft}dt$$
The total energy of each time $t_{j}$ is $E_{t_{j}}$:
(5)$$E_{t_{j}}=\sum_{i=1}^{N}e\left(f_{i},t_{j}\right)$$
(6)$$e\left(f_{i,}t_{j}\right)={\left|H\left(f_{i,}t_{j}\right)\right|}^{2}$$
where $e\left(f_{i},t_{j}\right)$ represents the energy of the *i*-th frequency point at the *j*-th time, $H\left(f_{i},t_{j}\right)$ represents the Generalized S-transform coefficient of the *i*-th frequency point at the *j*-th time, and then the percent of the energy of the *i*-th frequency point is:
(7)$$p\left(f_{i},t_{j}\right)=\frac{e\left(f_{i},t_{j}\right)}{E_{t_{j}}},i=1,2,\ldots ,N$$


In which $p\left(f_{i},t_{j}\right)$ is the percent of the energy of the *i*-th frequency point $f_{i}$ at the *j*-th time in the whole signal energy $E_{t_{j}}$ and $\sum_{i=1}^{N}p\left(f_{i},t_{j}\right)=1$.

So, the time–frequency entropy based on Generalized S-transform obtained by the follow formula:
(8)$$S_{t_{j}}=-\sum_{i=1}^{N}p\left(f_{i},t_{j}\right)\;ln\;\left(p\left(f_{i},t_{j}\right)\right)$$


### 2.2. MPE

Multi-scale permutation entropy is essentially the permutation entropy of time series using different scale factors. The specific steps of the calculation about MPE are as follows:

Step 1. For embedding dimension m, time delay τ, the time series X={x(1), x(2),…,x(N)} can be reconstructed in phase space as
(9)$$X^{m,\tau }=\{X^{m,\tau }\left(1\right),X^{m,\tau }\left(2\right),\ldots ,X^{m,\tau }\left(k\right)\ldots ,X^{m,\tau }\left(N-\left(m-1\right)\tau \right)\}$$
$X^{m,\tau }\left(k\right)$ can be expressed as
(10)$$X^{m,\tau }\left(k\right)=\{x\left(k\right),x\left(k+\tau \right),\ldots ,x\left(k+\left(m-1\right)\tau \right)\}$$
where k=1,2,…,N−(m−1)τ.

Step 2. {x(k),x(k+τ),…,x(k+(m−1)τ)} in an ascending order can be rearranged as $\{x\left(k+\left(\nu_{1}-1\right)\tau \right)\leq x\left(k+\left(\nu_{2}-1\right)\tau \right)\leq \cdots \leq x\left(k+\left(\nu_{2}-1\right)\tau \right)\}$, and the symbol index sequence $\pi_{l}^{m,\tau }$ can be obtained. It is clear that $\pi_{l}^{m,\tau }$ have m! possible values which can be expressed as
(11)$$\pi_{l}^{m,\tau }=\left\{\nu_{1},\nu_{2},\cdots ,\nu_{m}\right\}$$


Step 3. $p\left(\pi_{l}^{m,\tau }\right)$ is defined as
(12)$$p\left(\pi_{l}^{m,\tau }\right)=\frac{∥\{k|k=1,\ldots ,N-\left(m-1\right)\tau ;X_{l}^{m,\tau }has\pi_{l}^{m,\tau }type\}∥}{N-\left(m-1\right)\tau }$$


Step 4. PE can be defined as
(13)$$PE\left(X,m,\tau \right)=-\sum_{l:\pi_{l}^{m,\tau }\in \Pi }p\left(\pi_{l}^{m,\tau }\right)\;ln\;\left(p\left(\pi_{l}^{m,\tau }\right)\right)$$


Step 5. For X={x(1), x(2),…,x(N)}, the coarse-grained time series $y^{s}\left(j\right)$ can be expressed as
(14)$$y^{s}\left(j\right)=\frac{1}{s}\sum_{i=\left(j-1\right)s+1}^{js}x\left(i\right)$$
where j=1,2, …, [N/s].

Step 6. MPE can be defined as
(15)$$MPE\left(X,m,\tau ,s\right)=PE\left(y^{s},m,\tau \right)$$


### 2.3. RCMWPE

MPE ignores the amplitude difference between the same permutation patterns and does not include the amplitude information of the time series. As the scale factor increases, the length of the time series becomes shorter, which results in a sudden change for MPE. To solve this problem, inspired by Fadlallah et al. [31] and Wu et al. [34], Refined composite multi-scale weighted permutation entropy can be proposed. The specific steps of the calculation about RCMWPE are as follows:

Step 1. $p_{\omega }\left(\pi_{l}^{m,\tau }\right)$ can be defined as
(16)$$p_{\omega }\left(\pi_{l}^{m,\tau }\right)=\frac{∥\{k|k=1,\ldots ,N-\left(m-1\right)\tau ;X_{l}^{m,\tau }has\pi_{l}^{m,\tau }type\}∥\omega_{k}}{\{N-\left(m-1\right)\tau \}\omega_{k}}$$
where $\omega_{k}$ can be expressed as
(17)$$\omega_{k}=\frac{1}{m}\sum_{q=1}^{m}{\left[x\left(k+\left(q-1\right)\tau \right)-{\bar{X}}^{m,\tau }\left(k\right)\right]}^{2}$$
where ${\bar{X}}^{m,\tau }\left(k\right)=\frac{1}{m}\sum_{q=1}^{m}\left[x\left(k+\left(q-1\right)\tau \right)\right]$. Then WPE(X,m,τ) can be defined as
(18)$$WPE\left(X,m,\tau \right)=-\sum_{l:\pi_{l}^{m,\tau }\in \Pi }p_{\omega }\left(\pi_{l}^{m,\tau }\right)\;ln\;\left(p_{\omega }\left(\pi_{l}^{m,\tau }\right)\right)$$


Step 2. For X={x(1), x(2),…,x(N)}, the coarse-grained time series $y^{s,q}\left(j\right)$ can be expressed as
(19)$$y^{s,q}\left(j\right)=\frac{1}{s}\sum_{i=\left(j-1\right)s+1}^{js+q-1}x\left(i\right)$$
where j=1,2, …, [(N+1)/s]−1,q=1,2,⋯,s.

MWPE(X,m,τ,s) can be defined as
(20)$$MWPE\left(X,m,\tau ,s\right)=WPE\left(y^{s,q},m,\tau \right)$$


Step 3. RCWPE(X,m,τ) is defined as
(21)$$CWPE\left(X,m,\tau \right)=-\frac{1}{s}\sum_{q=1}^{s}\sum_{l:\pi_{l}^{m,\tau }\in \Pi }\bar{p_{\omega }}\left(\pi_{l}^{m,\tau }\right)\;ln\;\left(\bar{p_{\omega }}\left(\pi_{l}^{m,\tau }\right)\right)$$
where $\bar{p_{\omega }}\left(\pi_{l}^{m,\tau }\right)=\frac{1}{s}\sum_{q=1}^{s}p_{\omega }\left(\pi_{l}^{m,\tau }\right)\;ln\;\left(p_{\omega }\left(\pi_{l}^{m,\tau }\right)\right)$, so we review RCMWPE(X,m,τ,s) as follows
(22)$$RCMWPE\left(X,m,\tau ,s\right)=RCWPE\left(y^{s,q},m,\tau \right)$$


### 2.4. Intra-Class Distance and Inter-Class Distance

The intra-class distance and the inter-class distance are widely used in feature selection as indicators of separability and compactness. The smaller the intra-class distance, the better the compactness of feature. The greater the inter-class distance, the better the separability of feature. 

For sample sets for each pattern in n -dimensional space {a(i)|i=1,2…,k}. Intra-class distance can be defined as
(23)$$D_{\mathrm{int}ra}=\sqrt{\frac{1}{k}\sum_{j=1}^{k}\left[\frac{1}{k-1}\sum_{i=1,i\neq j}^{k}\sum_{k=1}^{n}{\left(a_{k}^{j}-a_{k}^{i}\right)}^{2}\right]}$$


The inter-class distance can be defined as
(24)$$D_{\mathrm{int}er}=\sqrt{\sum_{k=1}^{n}{\left(m_{1_{k}}-m_{2_{k}}\right)}^{2}}$$
where $m_{1_{k}}$ and $m_{2_{k}}$ are the *k*-th component of the two types of pattern sample sets. In feature selection, the intra-class distance should be as small as possible and the inter-class distance should be as large as possible.

## 3. Experimental Methods and Results

### 3.1. Experimental System

The experimental system is shown in Figure 1. The 15 groups of experimental porcine muscle tissues were fresh. The povidone and 95% alcohol were mixed with 1:4 proportion, and mixing them and water with 1:20 proportion to remove oxygen in water, waiting for about one hour before experiment. The porcine muscle tissue was mounted on a rubber board and immersed in water under HIFU source (PRO2008, Shenzhen, China). The 3D position system was used to control the position of transducer through computer. In the experiment, the HIFU source was self-focusing transducer with concave spherical surface and the circular hole at the top allowed the B-mode ultrasound probe to pass through. The center frequency of the B-mode ultrasound probe was 3.5 MHz. The center frequency of HIFU transducer was 1.39 MHz and the input electric power of transducer can be adjusted during HIFU experiment. The porcine muscles were irradiated by HIFU. The treatment process was monitored and the ultrasonic echo signals were obtained by B-mode ultrasound probe. The ultrasonic echo signals were converted into digital signals for preservation by a digital oscilloscope (Model MDO3032, Tektronix, USA). The temperature of porcine muscle tissue in treatment region was estimated by the thermocouple (DT-3891G, Shenzhen, China) which was fixed in the treatment region. Finally, the actual denatured status of biological tissues were determined by sliced biological tissue. In the experiments, 300 ultrasonic echo signals were collected from 15 groups of porcine muscle samples, including 150 cases in normal status and 150 cases in denatured status. Figure 2 show the pictures of sliced porcine muscle tissue in normal and denatured status.

### 3.2. Interception of Ultrasonic Scattered Echo Signal Based on Time-Frequency Entropy

Time-frequency entropy based on GST is used for intercepting ultrasonic scattered echo signals from HIFU treatment region. Figure 3a is the ultrasonic echo signal from the HIFU-irradiated porcine muscle tissue (including normal region and treatment region), and the center frequency of the ultrasonic echo signal is 3.5 MHz. Then, the time-frequency spectrum of the signal is obtained by GST. As shown as Figure 3b, it is clearly observed that the time-frequency spectrum well locates the distribution of the ultrasonic echo signal. The time-frequency spectrum energy of the A and C segments (normal region) mainly concentrate near 3.5 MHz, and the time-frequency spectrum energy of the B segment (treatment region) mainly concentrates below 3.5 MHz. This is because the temperature of treatment region increases after being irradiated by HIFU. This results in the greater attenuation and frequency shift of the ultrasonic echo signal from the treatment region. Then, time-frequency entropy of ultrasonic echo signal, calculated by the proposed method, is shown in Figure 4. It can be observed that the time-frequency entropy has a peak when ultrasonic signal travels across the boundary between normal region and treatment region in tissues.

Therefore, it is confirmed that the time-frequency entropy based on GST can be used for intercepting ultrasonic scattered echo signals and Figure 5 shows the intercepted result. The solid line in Figure 5 is the starting point of ultrasonic scattered echo signals, and the dashed line is the end-point. The number of sampling points of the intercepted ultrasonic scattered echo signal is 453, the signal sampling frequency is 20 MHz and the acoustic velocity defaults as 1550 m/s. The depth of theoretical treatment region can be calculated as 17.6 mm, As shown in Figure 2b, the depth of actual treatment region is 16–19 mm. This proves that the proposed method is reliable.

### 3.3. Comparison between Different Entropies of Simulated Signal

To prove the advantage of the RCMWPE method, the white noise with 5000 data points are generated, and we analyze the MPE, MWPE and RCMWPE results on white noise. According to the previous reports [29,33], we select embedding dimension m = 3, 4, 5, 6, 7, time delay τ = 2. The results of different entropies of white noise using scale factor 1–50 are shown in Figure 6.

As shown in Figure 6, while the embedding dimension m increases, all entropy values decrease. It is clearly observed that when m = 3,4,5, the declines of MPE, MWPE and RCMWPE are not evident with the increase of scale factor, and the superiority of multi-scale analysis cannot be displayed effectively. When m = 6,7, MPE, MWPE and RCMWPE all show a significant decreasing trend as scale factor increases. In addition, the values of MWPE and RCMWPE decrease faster than that of MPE, indicating that MWPE and RCMWPE can be more sensitive to extracting time series complexity including amplitude information. Furthermore, compared with MWPE and MPE, RCMWPE can reduce the entropy value fluctuation, which demonstrates the stability and reliability of RCMWPE. In general, RCMWPE not only measures the complexity of time series including amplitude information with multi-scales, but also improves the stability of multi-scale entropy.

### 3.4. MPE and RCMWPE of Actual Ultrasonic Scattered Echo Signals 

300 ultrasonic scattered echo signals (including 150 cases from normal tissues and 150 cases from denatured tissues) are analyzed by MPE and RCMWPE. According to [30], the embedding dimension is selected as 7 and delay time is 2. The MPE and RCMWPE values of 300 ultrasonic scattered echo signals from normal and denatured tissues using scale factors 1–20 are shown in Figure 7. It is clearly observed that the MPE and RCMWPE values of ultrasonic scattered echo signals of denatured biological tissues are higher than normal biological tissues. Both MPE and RCMWPE can distinguish normal tissues and denatured tissues. However, there are fewer feature points in the overlap region between RCMWPE of denatured tissues and normal tissues compared with MPE, which indicates that RCMWPE has better separability than MPE in identification of denatured biological tissues.

### 3.5. Intra-Class Distance and Inter-Class Distance of the MPE and RCMWPE 

In order to further prove the advantage of RCMWPE, the intra-class distance and inter-class distance of the MPE and RCMWPE of the normal tissues and denatured tissues are calculated for various scale factors. Figure 8 show the intra-class distance and inter-class distance of MPE and RCMWPE using scale factors 1–20. From Figure 8, it can be clearly observed that the intra-class distance of RCMWPE is less than MPE, and the inter-class distance of RCMWPE is greater than MPE. This indicates that RCMWPE have better compactness and separability compared with MPE, and RCMWPE can better identify whether biological tissues have been denatured. At the same time, when the scale factor is 12, the intra-class distance of RCMWPE is minimum and the inter-class distance of RCMWPE is the largest.

## 4. Conclusions

This paper focuses on identification of denatured biological tissues during HIFU treatment. Time-frequency entropy based on GST can be used to accurately intercept ultrasonic scattered echo signals from HIFU treatment region. In addition, a novel method, named RCMWPE is put forward, the comparison results with MPE prove the superiority of RCMWPE method. The MPE and RCMWPE are applied to the actual ultrasonic scattered echo signals of two types of tissues. The results show that RCMWPE have better separability and stability to identify whether biological tissues are denatured compared with MPE. Furthermore, when the scale factor is selected as 12, the optimal identification effect of normal tissue and denatured tissue can be obtained in the process of RCMWPE analysis for ultrasonic signals. In the future research, the non-uniform embedding may be adopted as a potential way to improve the tissue classification problem.

## Figures and Tables

**Figure 1 entropy-21-00666-f001:**
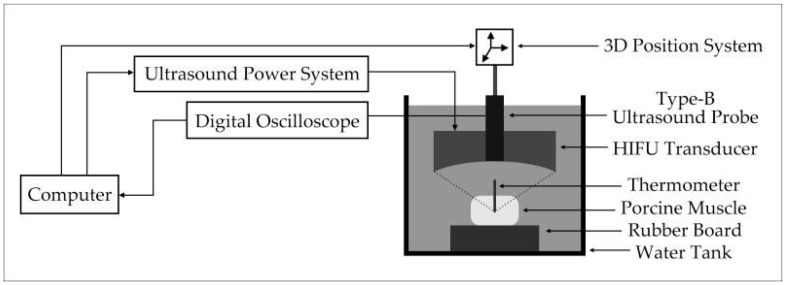
Experimental system.

**Figure 2 entropy-21-00666-f002:**
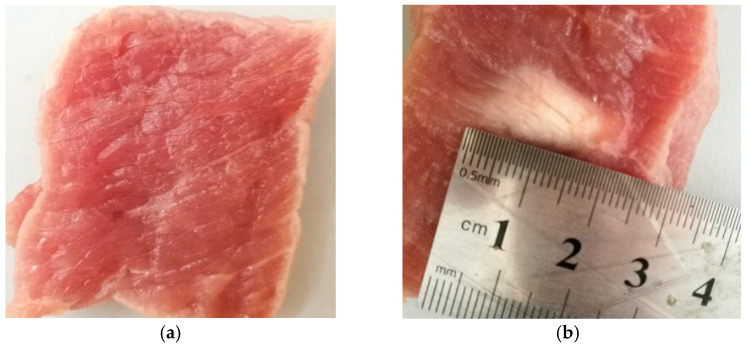
Pictures of sliced normal and denatured porcine muscle tissues. (**a**) normal tissue; (**b**)denatured tissue.

**Figure 3 entropy-21-00666-f003:**
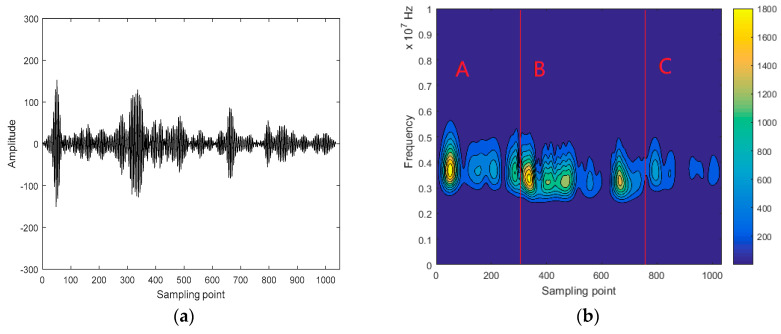
ultrasonic echo signal and its time-frequency spectrum. (**a**) The ultrasonic echo signal from the whole porcine muscle tissue; (**b**) Time-frequency spectrum obtained by GST.

**Figure 4 entropy-21-00666-f004:**
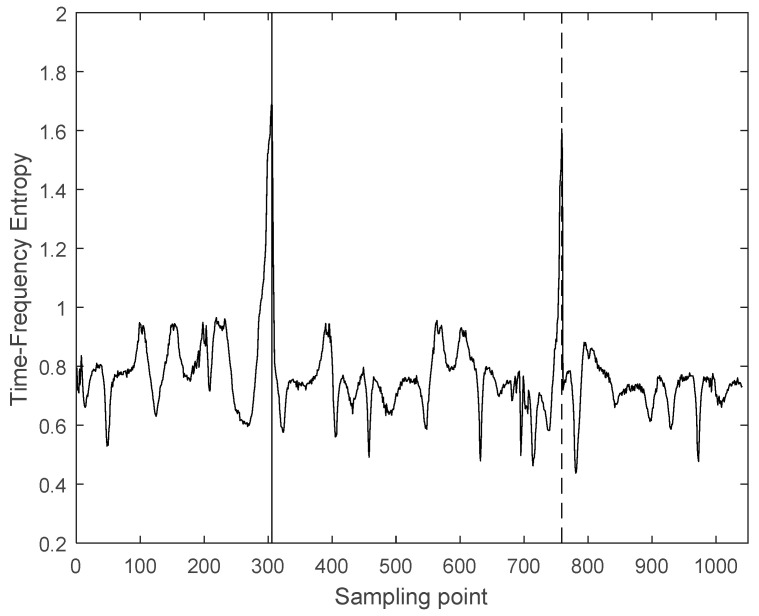
The time-frequency entropy of ultrasonic echo signal.

**Figure 5 entropy-21-00666-f005:**
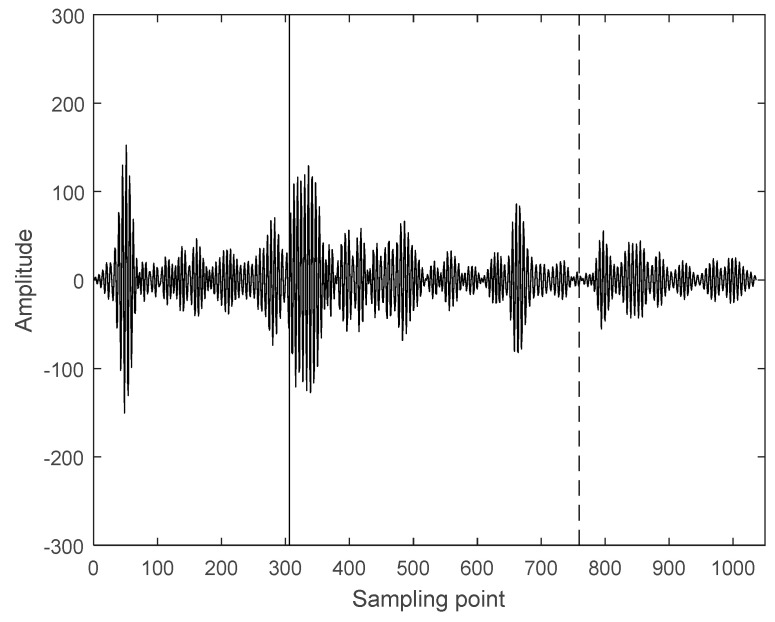
The intercepted result of the ultrasonic scattered echo signal from HIFU treatment region.

**Figure 6 entropy-21-00666-f006:**
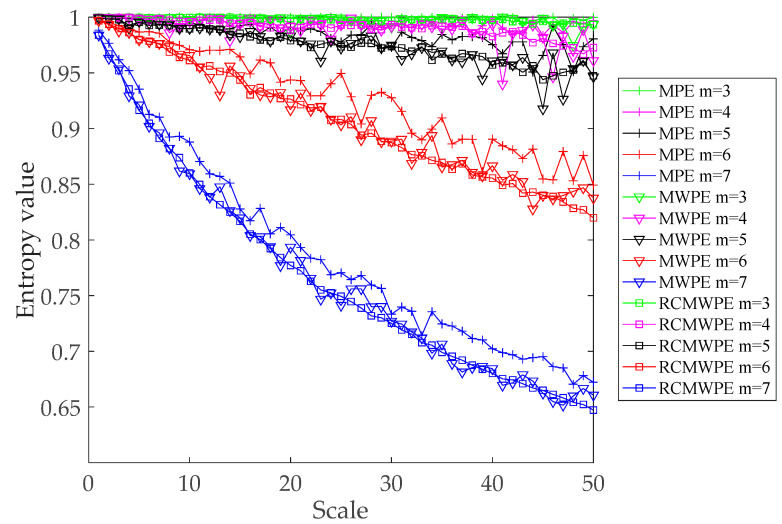
Different entropies values of the white noise using different scale factors.

**Figure 7 entropy-21-00666-f007:**
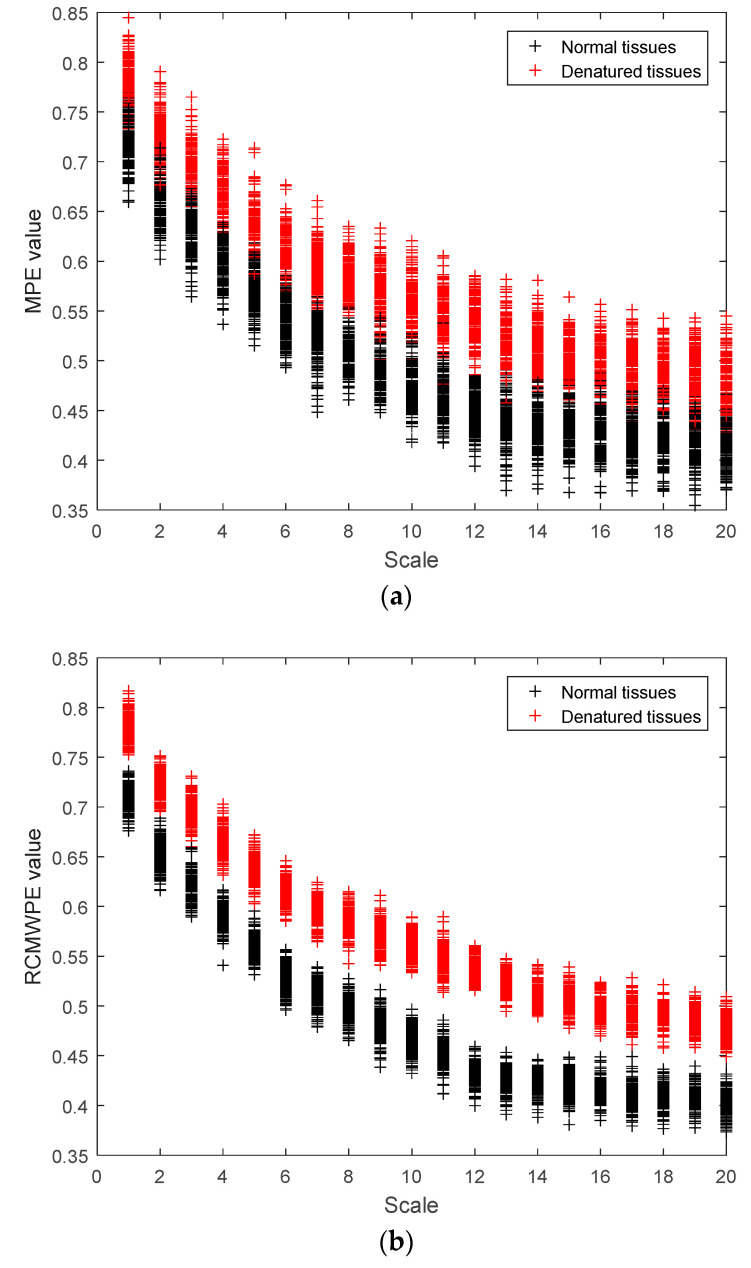
The MPE and RCMWPE distribution of normal and denatured tissues using different scale factors. (**a**) MPE; (**b**) RCMWPE.

**Figure 8 entropy-21-00666-f008:**
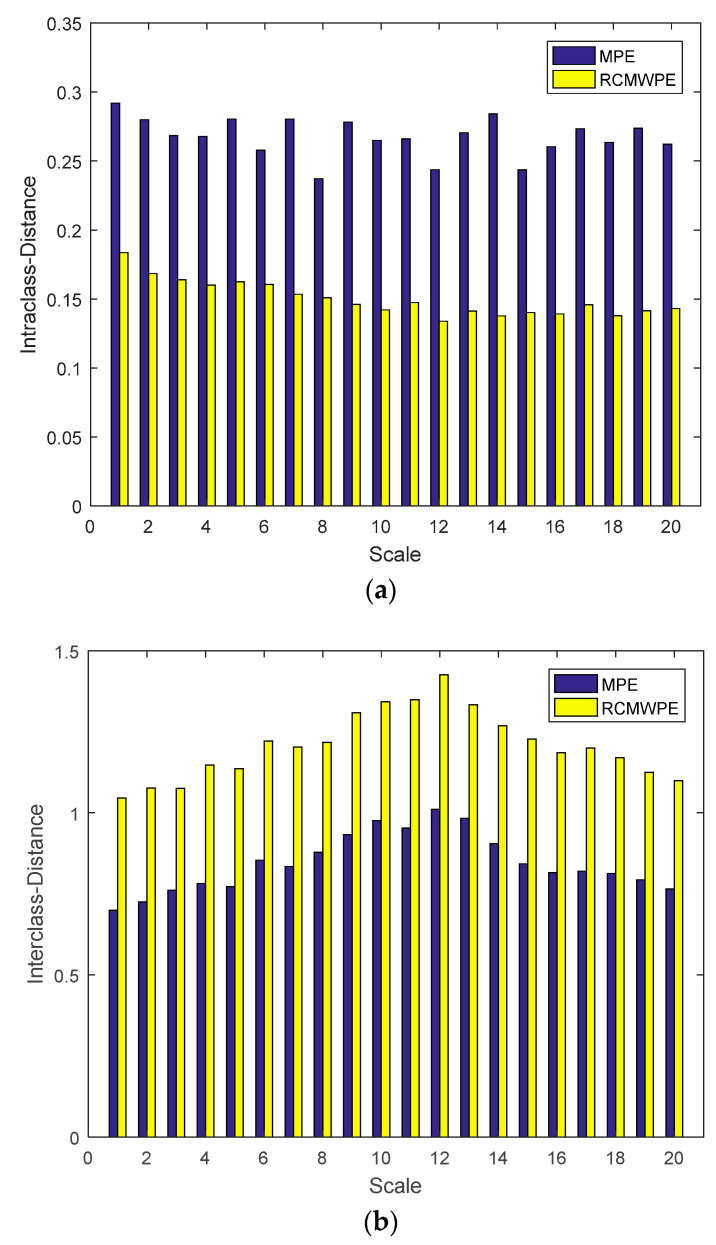
intra-class distance and inter-class distance of MPE and RCMWPE using different scale factors. (**a**) Intra-class distance of MPE and RCMWPE using different scale factors; (**b**) Inter-class distance of MPE and RCMWPE using different scale factors.

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
