# Peer review of "Identification of Denatured Biological Tissues Based on Time-Frequency Entropy and Refined Composite Multi-Scale Weighted Permutation Entropy during HIFU Treatment"

_entropy, 2019, doi:10.3390/e21070666_

Round 1

Reviewer 1 Report

In my opinion, the paper is of some interest. It suggests that a variant of permutation entropy (MPE), called refined composite multi-scale weighted permutation entropy (RCMWPE), can be used for identifying denatured biological tissue. The results justify publication in Entropy, there are however some points with should be improved:

The representation the Generalized S-transform is not perfect, in particular some definition of the energy seems to be missing.

A better motivation of the used entropy is mandatory. In particular, it should be explained why the RCMWPE was introduced.

I propose also to include (weighted permutation entropy) WPE into the comparision of methods because it is a prestep to RCMWPE.

Author Response

Dear Reviewer:

Thank you for your comments concerning our manuscript entitled”Identification of denatured biological tissues based on time-frequency entropy and refined composite multi-scale weighted permutation entropy during HIFU treatment”(ID: entropy- 542041-Minor Revisions). Those comments are all valuable and very helpful for revising and improving our paper, as well as the important guiding significance to our researches. We have studied comments carefully and have made correction which we hope meet with approval. Revised portion are marked in red in the paper. The main corrections in the paper and the responds to the reviewer’s comments are as flowing:

1. The representation the Generalized S-transform is not perfect, in particular some definition of the energy seems to be missing.

Response: We have made correction according to the Reviewer’s comments. The definition of energy has been added in this paper.

2. A better motivation of the used entropy is mandatory. In particular, it should be explained why the RCMWPE was introduced.

Response: Considering the Reviewer’s suggestion, we have made the corresponding correction in the introduction to explain the advantages of RCMWPE.

3. I propose also to include (weighted permutation entropy) WPE into the comparision of methods because it is a prestep to RCMWPE.

Response: We have made correction according to the Reviewer’s suggestion. Comparision of MWPE was added in section 3.3 to prove further the advantages of RCMWPE .

Once againthank you very much for your comments and suggestions.

Reviewer 2 Report

In general, this is an interesting approach to a classical problem. Thousands of efforts has been spent during the last decades for the identification and classification of biological tissues. This is another approach - and the presented results seems promising. 

The only comment from my side is in regards to the mathematical techniques used in the signal embedding process. Authors so employ the classical uniform embedding (all time delays are equal). It has been demonstrated in several recent publications that non-uniform embedding (when all time delays are not necessarily equal) may yield better results compared to uniform embedding. 

A typical result in non-uniform embedding (directly related to entropy) is published in: Permutation entropy based on non-uniform embedding. Entropy (2018) vol. 20(8), article ID: 612.

Authors are requested at least to comment this article. A potential for using non-uniform embedding in the presented tissue classification problem could be at least a good objective for further research. 

Author Response

Dear Reviewer:

Thank you for your comments concerning our manuscript entitled”Identification of denatured biological tissues based on time-frequency entropy and refined composite multi-scale weighted permutation entropy during HIFU treatment”(ID: entropy- 542041-Minor Revisions). Those comments are all valuable and very helpful for revising and improving our paper, as well as the important guiding significance to our researches. We have studied comments carefully and have made correction which we hope meet with approval. Revised portion are marked in red in the paper. The main corrections in the paper and the responds to the reviewer’s comments are as flowing:

Comments: The only comment from my side is in regards to the mathematical techniques used in the signal embedding process. Authors so employ the classical uniform embedding (all time delays are equal). It has been demonstrated in several recent publications that non-uniform embedding (when all time delays are not necessarily equal) may yield better results compared to uniform embedding

Response: We have made correction according to the Reviewer’s comment. The article “Permutation entropy based on non-uniform embedding,Entropy (2018) vol. 20(8)” has been added in the references.

Once againthank you very much for your comments and suggestions.

Entropy EISSN 1099-4300 Published by MDPI AG, Basel, Switzerland RSS E-Mail Table of Contents Alert
Back to Top